# Flexible Wearable Strain Sensors Based on Laser-Induced Graphene for Monitoring Human Physiological Signals

**DOI:** 10.3390/polym15173553

**Published:** 2023-08-26

**Authors:** Yao Zou, Mian Zhong, Shichen Li, Zehao Qing, Xiaoqing Xing, Guochong Gong, Ran Yan, Wenfeng Qin, Jiaqing Shen, Huazhong Zhang, Yong Jiang, Zhenhua Wang, Chao Zhou

**Affiliations:** 1Institute of Electronic and Electrical Engineering, Civil Aviation Flight University of China, Deyang 618307, China; zouyao670427@163.com (Y.Z.); lscxy19990217@163.com (S.L.); rekinleypa@163.com (Z.Q.); tezrj.love@163.com (X.X.); shengjiaqing@cafuc.edu.cn (J.S.); zhz_233@yeah.net (H.Z.); zc_cafuc@163.com (C.Z.); 2Institute of Civil Aviation Intelligent Sensing and Advanced Detection Technology, Civil Aviation Flight University of China, Deyang 618307, China; 3College of Aviation Engineering, Civil Aviation Flight University of China, Deyang 618307, China; ggc1261@163.com (G.G.); yanran331@163.com (R.Y.); qwfgrh@126.com (W.Q.); 4School of Mathematics and Physics, Southwest University of Science and Technology, Mianyang 621010, China; y_jiang@swust.edu.cn; 5Institute of Electronic and Electrical Engineering, Northwestern Polytechnical University, Xi’an 710129, China

**Keywords:** flexible wearable strain sensor, laser-induced graphene (LIG), polyimide (PI) film, physiological signals, health monitoring

## Abstract

Flexible wearable strain sensors based on laser-induced graphene (LIG) have attracted significant interest due to their simple preparation process, three-dimensional porous structure, excellent electromechanical characteristics, and remarkable mechanical robustness. In this study, we demonstrated that LIG with various defects could be prepared on the surface of polyimide (PI) film, patterned in a single step by adjusting the scanning speed while maintaining a constant laser power of 12.4 W, and subjected to two repeated scans under ambient air conditions. The results indicated that LIG produced at a scanning speed of 70 mm/s exhibited an obvious stacked honeycomb micropore structure, and the flexible strain sensor fabricated with this material demonstrated stable resistance. The sensor exhibited high sensitivity within a low strain range of 0.4–8.0%, with the gauge factor (GF) reaching 107.8. The sensor demonstrated excellent stability and repeatable response at a strain of 2% after approximately 1000 repetitions. The flexible wearable LIG-based sensor with a serpentine bending structure could be used to detect various physiological signals, including pulse, finger bending, back of the hand relaxation and gripping, blinking eyes, smiling, drinking water, and speaking. The results of this study may serve as a reference for future applications in health monitoring, medical rehabilitation, and human–computer interactions.

## 1. Introduction

With increasing emphasis on health and quality of life, wearable technology has rapidly emerged as a compelling and innovative solution. Flexible sensors play a critical role as core components in wearable devices due to their excellent flexibility, malleability, natural fit with the human skin surface, real-time monitoring, and the collection of physiological signal data. These sensors have demonstrated significant potential in health monitoring, medical rehabilitation, smart wearable devices, human–computer interaction, and other fields [1,2]. Conductive materials such as AgNWs [3], PEDOT [4], MXenes [5], and graphene [6] have been commonly used in the research on flexible sensors. However, these materials have complex preparation processes, resulting in high costs and long preparation cycles.

Laser-induced graphene (LIG) [7] has received considerable attention in recent years as a promising material for flexible, high-performance, and cost-effective sensors [8]. LIG can be produced through photothermal or photochemical reactions on carbon precursors directly irradiated by a laser in air, resulting in a three-dimensional (3D) porous structure with an exceptional surface area, scalability, and stability [9,10]. This approach offers single-step in situ fabrication and pattern customization of flexible sensors [11,12], offering high sensitivity, excellent biocompatibility, and robustness [13]. LIG can be synthesized using diverse carbon precursors, including renewable sources such as trees [14], paper [15], and cloth [16], as well as high-performance polymers such as polyetherimide (PEI) [17], phenolic resin (PR) [18], polydimethylsiloxane (PDMS) [19], poly-ether-ether-ketone (PEEK) [20] and polyimide (PI) [21]. Therefore, LIG is considered as one of the best methods to produce high-performance flexible wearable sensors [22], and LIG-based flexible wearable sensors offer promising market prospects in various fields, including smart wearable electronic devices, medical rehabilitation, and scientific exercises [23,24,25]. Chen et al. [26] prepared LIG on the surface of PI with square, zigzag, and shutter patterns, where shutter-patterned E-skins were able to be used to detect various physiological signals and had an alarm function when an abnormal situation was detected. Raza et al. [27] prepared flexible strain sensors and flexible pressure sensors by preparing LIGs on the PI surface and transferring them to the PDMS surface, which can be used for monitoring during volleyball training. Vi’centic´ et al. [28] prepared LIG by preparing it on the PI surface and transferring it to the PDMS surface, which was affixed to the median vein position of the elbow for detecting the heartbeat of a person. Yang et al. [8] used laser irradiation on the surface of the PI coated with HfSe_2_ to generate HfSe_2_-modified LIG nanosheets with high sensitivity and linearity that can be used to monitor various limb movements.

Although a number of studies on LIG morphology, structure, properties and applications have existed, the preparation of flexible wearables by preparing LIG remains a challenge from the perspective of parameter regulation, material selection, and device selection during the preparation process [29,30,31]. The performance of LIG-based flexible wearable sensors, including their sensitivity, response time, flexibility, deformability, preparation efficiency, and cost, will be significantly influenced by these factors [32]. Luo et al. [33] investigated the effects of laser power and laser scanning speed on the morphology of LIG and the electrical performance of LIG sensors, and it was found that by varying the laser power or laser scanning speed, the sensitivity of the sensors would be affected very significantly. Huang et al. [34] prepared wave-shaped LIG sensors and planar-shaped LIG sensors, and it was found that wave-shaped LIG sensors had higher sensitivity. Santos et al. [30] prepared LIG using two devices—infrared lasers and ultraviolet lasers—and it was found that the UV laser has a higher inscription resolution, but the IR laser prepared sensors with higher sensitivity.

In order to address the existing challenges, the preparation process needs to be explored and optimized to enhance efficiency, reduce costs, and further improve sensor performance, this study specifically examined the impact of laser scanning speed on LIG morphology and performance. A series of LIG samples with various morphological and structural characteristics was prepared using a pulsed CO_2_ infrared laser (Synrad P150, Novanta Inc., Bedford, MA, USA) with a wavelength of 10.64 μm, a power output of 12.4 W, with two scans per sample, and a pulse repetition rate of 20 kHz. The sample pattern was uniquely designed with a serpentine bending structure to enhance the flexibility and deformability of the LIG [35]. After initial characterization and performance testing, a flexible strain sensor with comprehensive performance was selected for detecting small strains in human physiology such as pulse and blinking of the eyes, or small amplitude physiological movements such as finger bending and hand relaxation/clenching.

## 2. Materials and Methods

### 2.1. Materials and Preparation Procedure

In this study, a CO_2_ infrared laser with a wavelength of 10.64 μm was directly applied onto thin PI film surfaces to prepare LIG with a porous network structure. The experimental procedure involved placing 2.5 cm × 2.5 cm commercial PI film with a thickness of 125 μm in an ultrasonic cleaning machine (Chun Rain Inc., Shenzhen, China). The film was then cleaned with deionized water at room temperature for 10 min and allowed to dry naturally on a stainless steel drainage plate. The next step involved affixing the PI film onto a stainless steel plate using polyethylene terephthalate (PET) (Wing Tai Co., Zhongshan, China) tape and securing the plate onto a high-precision moving platform in three coordinates. Then, a laser was used to directly inscribe the LIG pattern of the serpentine curved structure onto the PI film surface, with a pattern size of 1.5 cm × 1.5 cm. Copper wire was fixed with conductive silver paste at the designated pin positions using this pattern. After curing, the LIG pattern was encapsulated with 50-µm-thick PI (Shenzhen Jihongda Plastic Products Co., Shenzhen, China) tape, as shown in Figure 1. The utilized laser consisted of a pulsed CO_2_ infrared laser operating at a power of 12.4 W, with two scanning cycles at a repeated frequency of 20 kHz, and defocused by 495 mm. Various scanning speeds were used in this experiment (75, 105, 140, and 170 mm/s) to investigate the effect of scanning speed on both the surface morphology and properties of LIG.

### 2.2. Characterization

In this study, various characterization and testing methods were used to conduct a comprehensive analysis of the laser direct writing PI products. A focused ion beam scanning electron microscope (FIB-SEM, Thermo Scientific Helios 5 CX, Thermo Fisher Scientific Inc., Waltham, MA, USA) was used to directly observe the 3D surface morphology of the PI products, while an energy dispersive spectrometer (EDS, Thermo Scientific Helios 5 CX, Thermo Fisher Scientific Inc., Waltham, MA, USA) was used to detect the material elements on the LIG and PI surfaces. The crystal structures of the LIG powder and PI were characterized by X-ray diffraction (XRD, Rigaku Ultima IV, Rigaku Corporation, Akishima, Japan), with Cu target radiation. Wide-angle diffraction scanning was performed at 40 kV and 40 mA within the range of 10–60° and at a scanning rate of 5°/min. The functional groups in the LIG and PI molecules were detected by Fourier transform infrared spectroscopy (FTIR, Thermo field Nicolet iS5, Thermo Fisher Scientific Inc., Waltham, MA, USA) in the mid-infrared region using ATR mode. The molecular structure, chemical composition, and molecular vibration information of the LIG powder and PI were analyzed using a Renishaw inVia confocal micro-Raman spectrometer (Raman, Gloucestershire, UK) with an excitation wavelength of 532 nm and a wave number range of 1000–3000 cm^−1^. The chemical properties of both LIG and PI surfaces were investigated using FEI ESCALAB Xi+ X-ray Photoelectron Spectroscopy (XPS, Thermo Fisher Scientific Inc., Waltham, MA, USA) with an Al target.

## 3. Results and Discussion

### 3.1. Surface Morphology Characterization

The surface morphology, crystallinity, and elemental composite of LIG were analyzed by SEM, XRD, Raman spectroscopy, FTIR spectroscopy, XPS analysis, and EDS. According to theoretical analysis, when the PI surface was irradiated by the CO_2_ infrared laser, a photothermal reaction occurred, resulting in a local high temperature and lattice vibrations. This reaction resulted in the cleavage and recombination of chemical bonds between the C, N, and O atoms, leading to the conversion of sp^3^ carbon atom hybridization to sp^2^ carbon atom hybridization [36]. Additionally, the recombination of N and O atoms generated gas escape. Consequently, porous LIG structures were formed [37,38]. Figure 2 shows the products generated on the PI film surface through direct laser writing, along with the SEM characterization results. The preparation diagrams of samples (a)–(d) and their corresponding SEM images provided direct observations of LIG formation at various laser scanning speeds of 75, 105, 140, and 170 mm/s, respectively. The surface microscopy characterization conducted by SEM is also shown in the figure. Direct observation with the naked eye revealed that only samples (a) and (b) showed a continuous serpentine LIG structure, while sample (c) showed discontinuous patterns, and sample (d) presented minimal laser ablation without any LIG formation.

Subsequently, SEM analysis was performed on the corners, bending lines, and pin circles for each of the four samples. The SEM images obtained from sample (a) revealed a distinct 3D network structure in LIG, which was generated at a scanning speed of 70 mm/s. The LIG surface exhibited numerous micro-holes, bumps, and gullies, which were likely caused by laser spot stacking. This was due to the higher laser radiation energy at high pulse repetition rates and a low scanning speed, resulting in a larger spot repetition area. SEM analysis of sample (b) revealed that LIG generated at a scanning speed of 105 mm/s exhibited relatively flat surfaces with numerous micropores. By contrast, SEM observations of sample (c) showed that pore structures appeared primarily at the corners and bends of LIG generated at a laser scanning speed of 140 mm/s, though with narrower widths. The laser energy followed Gaussian distribution [31], with the highest energy concentrated at the center and gradually decreasing toward both sides. However, no discernible pore structures were observed in the circular region. This suggested that at this scanning speed, the laser energy at the corners and curves was sufficiently high to reach the threshold for generating LIG, while laser energy at the circles was insufficient for PI to undergo significant deformation and generate gas or amorphous carbon. According to the SEM observations of sample (d), we observed that only amorphous carbon and gas could be generated under this laser parameter, and LIG could not be generated. Therefore, samples (a) and (b) were subjected to further characterization and analysis at the aforementioned four distinct laser scanning speeds.

### 3.2. Surface Structure and Spectroscopy Characterization

To further analyze the generated LIG, the samples produced at laser scanning speeds of 70 and 105 mm/s were characterized as shown in Figure 3. Figure 3a shows the XRD analysis of both PI and LIG. Compared to PI, LIG produced at a scanning speed of 70 and 105 mm/s exhibited obvious (002) and (100) peaks, corresponding to diffraction peaks observed at angles of 25.18° and 43.56°, respectively. These two diffraction peaks corresponded to the interlayer structure and planar structure of LIG [39]. The Raman spectrum showed the distinctive features of LIG. Figure 3b illustrates the Raman spectra analysis of PI and LIG, revealing distinct differences in the peak patterns between the two materials. Specifically, three peaks were generated at various scanning speeds for LIG. In particular, the D peak (1346 cm^−1^), G peak (1584.9 cm^−1^), and 2D peak (2687.57 cm^−1^) were observed, which were consistent with previously reported data [40]. Subsequently, the peak intensities of the two line segments were compared. The Raman spectra of LIG obtained at a scanning speed of 105 mm/s had an I_D_/I_G_ ratio of 1.167, while the samples obtained at a scanning speed of 70 mm/s exhibited an I_D_/I_G_ ratio of 1.295, indicating that LIG produced at a lower scanning speed had fewer defects. By comparing the I_2D_/I_G_ ratio, we observed that the LIG sample produced at a scanning speed of 70 mm/s had a ratio of 0.774, while the LIG sample produced at a scanning speed of 105 mm/s had a ratio of 0.631. This indicated that the former had fewer layers and better quality. Therefore, according to the figure, we inferred that LIG generated at a scanning speed of 70 mm/s exhibited superior quality. Figure 3c illustrates the FTIR analysis results of PI and LIG. As indicated by the line segment analysis in the figure, both PI and LIG exhibited distinct absorption peaks within the range of 500–1800 cm^−1^, which suggested a significant reduction in C-O, C=O, and C-N bonds [41]. This finding confirmed that a photothermal reaction occurred during the laser direct writing process, forming sp^2^ carbon.

XPS analysis was performed on LIG and PI, as shown in Figure 3d, where the C peak of both samples was clearly visible. The results indicated that the carbon content values of PI was 78.22%, and the carbon content of the LIG produced at scanning speeds of 105 and 70 mm/s was 89.07% and 91.91%, respectively. The highest carbon content of LIG was obtained at a scanning speed of 70 mm/s, and the O peak was observed in the spectra. According to the figure, PI exhibited the highest intensity of the O peak, constituting 15.61% of the atomic content. The corresponding values for LIG generated at scanning speeds of 105 and 70 mm/s were 9.04% and 6.69%, respectively, indicating that LIG generated at a scanning speed of 70 mm/s exhibited the lowest O atom content value. Furthermore, the N atom was analyzed, and we observed that the N peak in PI exhibited greater prominence at 6.17%. At scanning speeds of 105 and 70 mm/s, the N peak was imperceptible to the naked eye, with atomic content values of 1.7% and 1.19%. This was due to the carbonization process of laser direct writing on PI, which converted the C atoms from sp3 hybrid to sp2 hybrid while also causing the O and N atoms to recombine into a gas that escaped. Peak splitting, fitting, and correction of C1s at this speed were performed using Avantage software, as shown in Figure 3e. The peaks of C1s were resolved into four components, namely, C=C, C-N, C-O, and C=O, with binding energies of 284.8, 285.48, 288.85, and 291.25 eV, respectively [42]. The O 1s spectrum shown in Figure 3f indicated significant noise due to a low signal-to-noise ratio. Compared to LIG generated by two groups with different scanning speeds in terms of atom content, we observed from Figure 3 that PI carbonization was more complete at a scanning speed of 70 mm/s.

Raman spectroscopy results in Figure 3b and XPS analysis results in Figure 3d indicated that the LIG generated at a scanning speed of 70 mm/s exhibited superior quality. Therefore, further detection and analysis of the sample were conducted as shown in Figure 4. Figure 4a shows the serpentine and curved pattern of the flexible sensor, and Figure 4b shows the SEM cross-section image of LIG in sample (a). As demonstrated in the diagram, the distance between the LIG surface and the bottom of PI was 257.9 μm, with a height of 168.9 μm for LIG and an original height of 125 μm for PI, indicating partial embedding of LIG within PI. Figure 3c shows a low magnification top view of sample (a), with LIG located on the left, while PI occupied the right. EDS detection revealed that the concentration of C elements was higher on the LIG side, as indicated by its darker color compared to that on the PI side. Conversely, O and N elements exhibited bright colors on the PI side, suggesting higher content in this region. However, due to low levels of N elements, their influence on image coloration was negligible [43].

### 3.3. Mechanical Performance

As shown in Figure 5, the electromechanical properties of the flexible strain sensors were evaluated by subjecting the samples to stretching and releasing, using a tensile testing machine (WNMC 400, Beijing, China). Real-time resistance variation detection of flexible sensors was performed using a digital multimeter (RIGOL DM3058E, RIGOL Technologies Co., Ltd., Suzhou, China). Figure 5a shows the relative resistance changes for the various strains conditions for sensors prepared using LIG generated by laser scanning speeds of 70 mm/s and 105 mm/s. The obtained data showed good fitting, with R^2^ values above 0.97, for the different strain ranges, indicating accurate measurements. The gauge factor (GF) was calculated by relating the relative resistance changes to the applied strains, as follows:(1)$$GF=\frac{\delta (R-R_{0})/R_{0}}{\delta \varepsilon }$$
where *GF* is the sensitivity factor of the sensor, *R* represents the real-time resistance, *R*_0_ is the initial resistance, (*R* − *R*_0_)/*R*_0_ indicates the relative resistance change of the sensor, and ε denotes the tensile strain of the sensor. As stretching increased, the relative resistance change also increased, showing a positive correlation between the two parameters. As shown in Figure 5b, within the strain range of 0.0–0.8% (low strain), the GF of the sensor corresponding to 70 mm/s is 18.4, and that of the sensor corresponding to 105 mm/s is 16.2. Within this strain range, the sensor experienced crack formation during the stretching and releasing process, resulting in a change in relative resistance. However, the cracks were small, and upon release, the stability of the sensor could be restored by establishing new conduction paths. Within the strain range of 0.8–5.6%, the GF of the sensor corresponding to 70 mm/s is 5.3, and that of the sensor corresponding to 105 mm/s is 1.5, and the LIG experienced slight fracture, while the conductive pathway was somewhat obstructed. In the strain range of 5.6–8%, the GF of the sensor corresponding to 70 mm/s reached 107.8, and that of the sensor corresponding to 105 mm/s is 4.5, indicating significant fracture of LIG during stretching, resulting in a reduction in the conductive pathway and a substantial change in relative resistance. Due to the higher sensitivity of the sensor corresponding to 70 mm/s, further performance analysis and applications will be conducted on this sensor.

Figure 5c shows a sensitivity comparison with relevant literature on LIG sensors, indicating that this sensor exhibits high sensitivity within the low strain range. Figure 5d shows the relative resistance changes for detecting a very small strain of 0.4%, indicating that the sensor had good stability and reliability for detecting small signals in 10 stretch–release cycles. As shown in Figure 5e, 10 detection cycles were carried out corresponding to 0.8%, 2.0%, 4.0%, 5.6%, 6.4%, and 7.2% stretch–release changes, indicating that the sensor had certain stability and reversibility at different strains. Figure 5f displays that the sensor has the same time response at tensile strains of 2.0%, 4.0%, 5.6%, and 6.4%, respectively. The sensor was tested for about 1000 cycles at 2% strain and showed good durability in Figure 5g. According to Figure 5h, the 3250–3300 s waveform in Figure 5g had good repeatability, indicating that the sensor could be used for the long-term monitoring of human physiological signals with good recoverability and reference.

### 3.4. Application Characterization

Due to the outstanding electromechanical properties of LIG, the flexible strain sensor fabricated with LIG exhibited a broad potential application range and could be used to detect physiological signals in various parts of the human body. Specifically, as shown in Figure 6, we focused on detecting physiological signals such as pulse, relaxation, gripping of the back of the hand, bending and straightening of the fingers, blinking of the eyes, smiling, drinking water, and speech, as flexible strain sensors have shown exceptional performance in these application scenarios. Due to their high sensitivity and repeatability, flexible strain sensors can precisely capture and measure even the slightest movements and physiological changes in the human body. For example, pulse detection can be used to monitor alterations in heart rate and blood pressure, and relaxation and grip detection on the back of the hand can be used to analyze hand movements and monitor rehabilitation. The detection of finger bending and straightening can be used for gesture recognition and control of hand movement. Additionally, detecting actions such as blinking, smiling, drinking water, and speaking may offer applications in areas such as emotion recognition, oral communication, and medical rehabilitation monitoring.

Physiological signals can be intricately linked to human health, and even minor fluctuations may have significant implications. Therefore, it is crucial for flexible strain sensors to detect subtle strains on the skin. In this experiment, we used flexible strain sensors to capture minute signals emanating from the human body such as blinking and smiling. Initially, we could detect a smile during our trial. As shown in Figure 6a, the flexible strain sensor was fixed to the face behind the corner of the mouth. Upon smiling, the skin on the face stretched and caused a disruption and subsequent reconnection of the conductive layer within the LIG sensor, leading to changes in the resistance values. Five distinct wave patterns were observed as a result. In addition, eye blinking was successfully detected, as shown in Figure 6b. The flexible strain sensor was attached to the posterior corner of the eye and the anterior temple to detect changes in resistance during eyelid opening and closure. After five tests, we observed a discernible pattern in the waveform of the relative resistance fluctuations.

The throat plays a vital role in the processes of swallowing and vocal exercises. By affixing a sensor to the throat, we could detect skin strain caused by swallowing or small strain resulting from laryngeal vibrations during speech, which altered the resistance value of the flexible strain sensor. Therefore, a flexible strain sensor was used for the first time to detect changes in resistance during laryngeal vocalization. The words “congratulations” and “hello” indicated significant resistance alterations, as shown in Figure 6c and Figure 6d, respectively. Furthermore, we observed variations in resistance when drinking water, as shown in Figure 6e, and the sensor exhibited regular fluctuations in its resistance value.

The hand serves as an indispensable tool in daily life and is used for lifting objects, typing, and performing various other tasks. Proper positioning of the hand and fingers during activities such as sports training, tool use, and yoga can provide more effective results and reduce the risk of long-term injury resulting from improper form or accidental trauma. In this study, a flexible strain sensor was used to measure the pulse of the wrist, as shown in Figure 6f. The resulting graph indicated that six pulses with similar waveforms were detected within a duration of 4.2 s, which fell within the normal range for adult pulse frequency [45]. The relaxation and contraction of the dorsal hand muscles, as well as the flexing and extension of the fingers, were subsequently detected. As shown in Figure 6g, the sensor returned to its original state with a low resistance value when the back of the hand was relaxed [46]. The principal component analysis presented in Figure 6h was analogous to that presented in Figure 6g. When the hand performed a gripped motion, the flexible strain sensor on the back of the hand deformed and caused reverse stretching of LIG along its position, resulting in increased distance between its internal conductive layers. During transition from relaxation to gripping on the back of the hand, disconnect occurred in portions of the conductive layer. As a result, the resistance value of the flexible strain sensor increased.

## 4. Conclusions

In this study, we used a CO_2_ infrared laser with a wavelength of 10.64 μm to directly write on a PI surface, resulting in the successful fabrication of serpentine curved structures, with four different laser scanning speeds. After testing and analyzing, the sensor prepared at a scanning speed of 70 mm/s exhibited high sensitivity within a low strain range of 0.4–8.0% (GF_max_ = 107.8). Additionally, the sensor demonstrated excellent stability and repeatable response at a strain of 2% after 1000 repetitions. The application test results demonstrated that the sensitivity of the flexible strain sensor was sufficient for detecting human physiological activity, including subtle wrist pulse strains and movements such as blinking, smiling, swallowing, talking, as well as hand relaxation/clenching and finger bending. The flexible strain sensor accurately detected corresponding signals with a certain degree of regularity and repeatability. In summary, flexible strain sensors based on LIG have significant potential in numerous fields, providing robust support for intelligent, personalized, and efficient health management, as well as sports training. With further research and development, LIG technology may be widely adopted and promoted in the future.

## Figures and Tables

**Figure 1 polymers-15-03553-f001:**
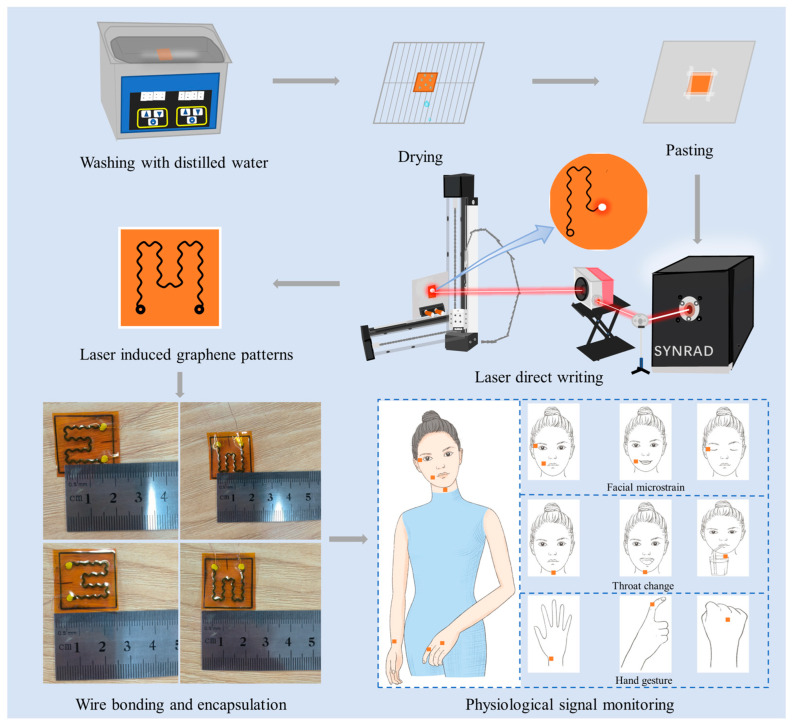
Fabrication procedure of the laser-induced graphene (LIG)-based flexible wearable sensor.

**Figure 2 polymers-15-03553-f002:**
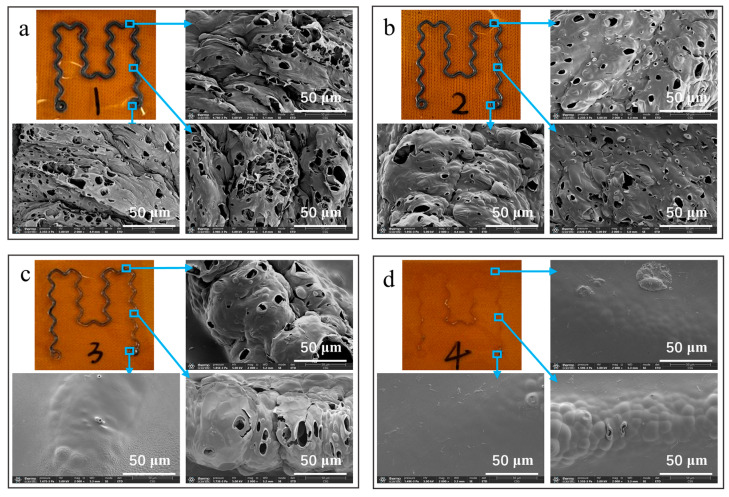
Scanning electron microscope (SEM) images of the LIG patterns obtained by using varying laser scanning speeds at the corners, curves, and circles: (**a**) 70 mm/s, (**b**) 105 mm/s, (**c**) 140 mm/s, and (**d**) 175 mm/s.

**Figure 3 polymers-15-03553-f003:**
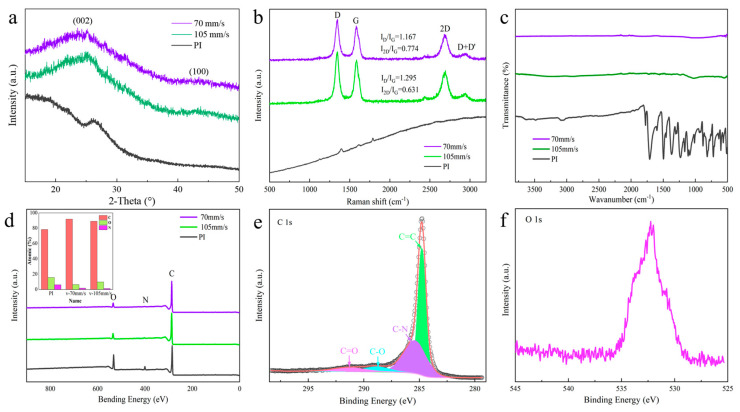
Characterization of the LIG powders prepared using varying laser scanning speeds: (**a**) XRD patterns showing the (002) peak at 25.18° and the (100) peak at 43.56°; (**b**) Raman spectroscopy showing the different forms of LIG, with peaks at the same locations; (**c**) FTIR analysis, indicating that LIG and PI possessed distinct functional groups; (**d**) XPS patterns indicating the positions of C, O, and N and their contents; (**e**) the split-peak fit of XPS C1s of LIG at a speed of 70 mm/s; and (**f**) peak pattern of O 1s.

**Figure 4 polymers-15-03553-f004:**
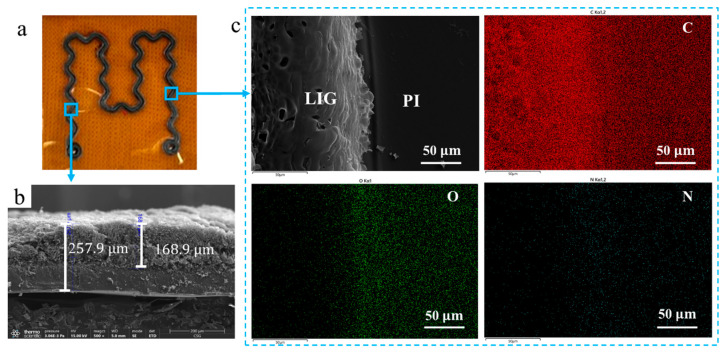
Characterization of LIG prepared with two scanning cycles: (**a**) sample diagram; (**b**) cross-sectional SEM images of the LIG film; (**c**) low-resolution top-view images of the LIG film and the corresponding SEM-EDS mapping images, presenting the carbon, oxygen, and nitrogen elements.

**Figure 5 polymers-15-03553-f005:**
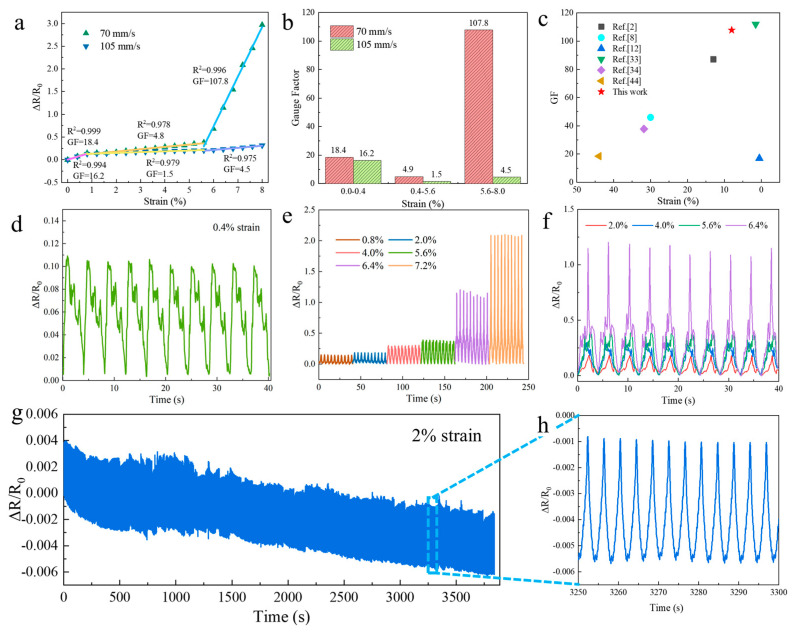
Performance tests conducted on the sensors: (**a**) strain sensitivity of sensors fabricated at two different speeds. (**b**) GF values across different strain range values for the sensor prepared at a speed of 70 and 105 mm/s; (**c**) comparisons of the existing LIG based sensors’ performance with the reported literature [44]; (**d**) sensors exhibiting the ability to detect strains as low as 0.4% with excellent stability; (**e**) relative resistance changes during 10 cycles of stretch-release for the sensor at small strains of 0.8%, 2.0%, 4.0%, 5.6%, 6.4%, and 7.2%; (**f**) time repeatability for different stretch lengths; (**g**) repeatability of the sensor demonstrated by approximately 1000 cycles at a strain of 2%; (**h**) stability of the waveform within the time range of 3250–3300 s during the cycle depicted in (**g**).

**Figure 6 polymers-15-03553-f006:**
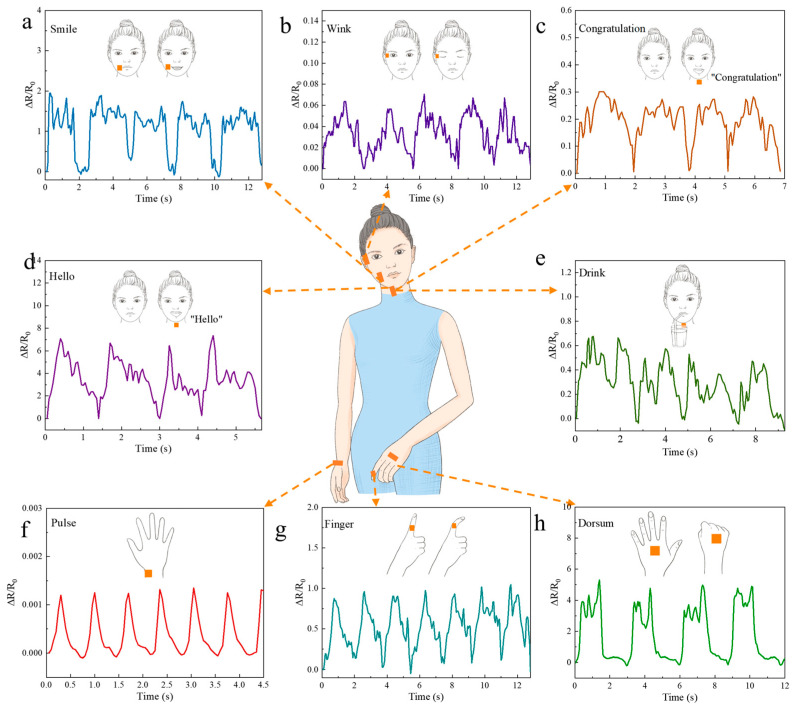
Application of the sensor for various small physiological signals: (**a**) application to the back of the mouth to test signal changes when smiling; (**b**) application to the corner of the eye to test changes when blinking; (**c**) application to the throat to test for signal changes when saying “congratulations”; (**d**) application to the throat to test signal changes when drinking water; (**e**) application to the throat to test signal changes when saying “hello”; (**f**) application to the wrist to test the changes in signal with a beating pulse; (**g**) application to the first knuckle of the index finger to test signal changes when the index finger was straightened and bent; (**h**) application to the back of the hand to test signal changes when the hand was extended and gripped.

## Data Availability

Data sharing is not applicable to this article.

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
