# Peer review of "Flexible Wearable Strain Sensors Based on Laser-Induced Graphene for Monitoring Human Physiological Signals"

_polymers, 2023, doi:10.3390/polym15173553_

Round 1
Reviewer 1 Report
Flexible wearable strain sensors based on laser–induced graphene for monitoring human physiological signals reports small information about the effect of different laser scanning speeds on LIG formation and its applications in sensors. Use of LIG for sensing applications is well-reported in the literature as mentioned by the author itself. Therefore, the scientific contribution from this study is very small, as the authors omitted two samples after SEM studies. Authors study have studied all four samples to have more scientific depth of the study. Therefore it can be considered after a significant revision with the inclusion of the below points
1. Authors should present at least 4 sample data, which form LIG with different laser scanning speeds.
2. Author should include different laser powers for a fixed scanning rate on the LIG film and it sensor performance.
3. Author should compare the existing LIG based sensors' performance with the reported literature (like a table).
4. Figure 1 scale bars are missing for photographs.
5. Figure 2 Scale bars and magnification details are completely blurred in the images (need clear SEM images)
6. Figure 3 should contain all samples data (all 4 scanning speeds or any four scanning speeds)
7. Figure 5 (e), (f) y axis needs axis units and numbers
8. Figure 6 (all panle) need Y axis units/numbers.
Reviewer 2 Report
The paper presented the application of graphene - based PI as a strain sensor with laser induced fabrication mechanism. The research work is properly introduced, starting from material characteristics and moving to application analysis. It needs minor revision before being processed further.
1- How did you measure the resistance? Is it using ohmmeter, or from I-V analysis of thin film using four probe station?
2- You mentioned that you used four different laser scanning speeds, but the results were focusing on only two speeds. Why?
3- Why did some results focus on only one speed, such as GF analysis? Also, is there an explanation for the low GF at certain strain range up to 5%?
4- A more detailed comparison with similar work in literature should be added, to show the contribution of this work.
N/A
Round 2
Reviewer 1 Report
Dear authors,
The revised manuscript can be accepted now.